# Using Radiomics and Machine Learning Applied to MRI to Predict Response to Neoadjuvant Chemotherapy in Locally Advanced Cervical Cancer

**DOI:** 10.3390/diagnostics13193139

**Published:** 2023-10-06

**Authors:** Valentina Chiappa, Giorgio Bogani, Matteo Interlenghi, Giulia Vittori Antisari, Christian Salvatore, Lucia Zanchi, Manuela Ludovisi, Umberto Leone Roberti Maggiore, Giuseppina Calareso, Edward Haeusler, Francesco Raspagliesi, Isabella Castiglioni

**Affiliations:** 1Gynecologic Oncology, Fondazione IRCCS Istituto Nazionale Tumori di Milano, 20133 Milan, Italy; giorgio.bogani@istitutotumori.mi.it (G.B.); umberto.leone@istitutotumori.mi.it (U.L.R.M.); francesco.raspagliesi@istitutotumori.mi.it (F.R.); 2DeepTrace Technologies S.R.L., 20126 Milan, Italy; interlenghi@deeptracetech.com (M.I.); salvatore@deeptracetech.com (C.S.); 3Azienda Ospedaliero-Universitaria di Verona, University of Verona, 37134 Verona, Italy; giulia.vittori@sacrocuore.it; 4Department of Science, Technology and Society, University School for Advanced Studies IUSS Pavia, 27100 Pavia, Italy; 5Department of Clinical, Surgical, Diagnostic and Pediatric Sciences, Unit of Obstetrics and Gynaecology, University of Pavia, IRCCS San Matteo Hospital Foundation, 27100 Pavia, Italy; lucia.zanchi@istitutotumori.mi.it; 6Department of Clinical Medicine, Life Health and Environmental Sciences, University of L’Aquila, 67100 L’Aquila, Italy; manuela.ludovisi@univaq.it; 7Radiology, Fondazione IRCCS Istituto Nazionale Tumori di Milano, 20133 Milan, Italy; giuseppina.calareso@istitutotumori.mi.it; 8Department of Anaesthesiology, Fondazione IRCCS Istituto Nazionale Tumori di Milano, 20133 Milan, Italy; edward.haeusler@istitutotumori.mi.it; 9Department of Physics G. Occhialini, University of Milan-Bicocca, 20133 Milan, Italy; isabella.castiglioni@unimib.it

**Keywords:** cervical cancer, MRI, radiomics, neoadjuvant chemotherapy, gynecology oncology

## Abstract

Neoadjuvant chemotherapy plus radical surgery could be a safe alternative to chemo-radiation in cervical cancer patients who are not willing to receive radiotherapy. The response to neoadjuvant chemotherapy is the main factor influencing the need for adjunctive treatments and survival. In the present paper we aim to develop a machine learning model based on cervix magnetic resonance imaging (MRI) images to stratify the single-subject risk of cervical cancer. We collected MRI images from 72 subjects. Among these subjects, 28 patients (38.9%) belonged to the “Not completely responding” class and 44 patients (61.1%) belonged to the ’Completely responding‘ class according to their response to treatment. This image set was used for the training and cross-validation of different machine learning models. A robust radiomic approach was applied, under the hypothesis that the radiomic features could be able to capture the disease heterogeneity among the two groups. Three models consisting of three ensembles of machine learning classifiers (random forests, support vector machines, and k-nearest neighbor classifiers) were developed for the binary classification task of interest (“Not completely responding” vs. “Completely responding”), based on supervised learning, using response to treatment as the reference standard. The best model showed an ROC-AUC (%) of 83 (majority vote), 82.3 (mean) [79.9–84.6], an accuracy (%) of 74, 74.1 [72.1–76.1], a sensitivity (%) of 71, 73.8 [68.7–78.9], and a specificity (%) of 75, 74.2 [71–77.5]. In conclusion, our preliminary data support the adoption of a radiomic-based approach to predict the response to neoadjuvant chemotherapy.

## 1. Introduction

Cervical cancer is the third most common gynecologic cancer and cause of death among gynecologic tumors in developed countries [1]. Although its incidence is decreasing in developed countries, in countries that do not have easy access to screening and prevention programs, cervical cancer remains a significant cause of cancer morbidity and mortality [1,2,3,4]. 

In early-stage disease, surgery is the mainstay of treatment, while in the locally advanced stage, chemoradiation is the most appropriate treatment. In women with locally advanced cervical cancer, guidelines suggest offering primary chemoradiation [5,6], although it is well known that the benefits of treatment are greater with an earlier (FIGO (International Federation of Gynecology and Obstetrics) stage IB2 to IIB) versus more advanced stage (FIGO stage III to IVA) [7,8,9]. 

The ESGO (European Society of Gynaecological Oncology)/ESTRO (European Society for Radiotherapy and Oncology)/ESP (European Society of Pathology) guidelines suggest the avoidance of tri-modality treatment in locally advanced cervical cancer (with negative lymph nodes on radiological staging). The treatment strategy should aim at avoiding the combination of radical surgery and external radiotherapy because of the significant increase in morbidity and no evident impact on overall survival (grade B). Definitive platinum-based chemoradiotherapy plus brachytherapy is the preferred treatment (grade A) [10]. Neoadjuvant chemotherapy (NACT) prior to radical hysterectomy does not offer an overall survival (OS) advantage compared to primary chemoradiation and, in randomized trials discussed below, it has been associated with worse disease-free survival (DFS). 

However, in parts of Europe, Asia, and South America, where access to radiotherapy is limited, neoadjuvant chemotherapy before radical hysterectomy can be an appropriate option for women with locally advanced disease. In a phase III trial, 633 women with stage IB2, IIA, or IIB squamous cervical cancer were randomly assigned to either three cycles of NACT (paclitaxel and carboplatin administered every three weeks) followed by a radical hysterectomy or standard chemoradiation. Compared with the standard chemoradiation, women receiving neoadjuvant chemotherapy followed by surgery experienced worse five-year DFS (69.3 versus 76.7 percent, respectively; hazard ratio [HR] 1.38, 95% CI 1.02–1.87) [11]. Five-year OS was similar between the two groups (75.4 versus 74.7 percent, respectively; HR 1.025, 95% CI 0.75–1.40) [11]. The preliminary results of a separate phase III trial (European Organisation for Research and Treatment of Cancer [EORTC] 55994), among 620 patients with stage IB2 to IIB cervical cancer showed that the patients randomly assigned to neoadjuvant chemotherapy followed by surgery experienced worsened five-year progression-free survival rates (57% versus 66%; difference 9 percent, 95% CI 2 to 18%) and comparable five-year OS rates (72% versus 76%; HR 0.87, 95% CI 0.65–0.15) to women receiving concomitant chemoradiation [12]. 

Historically, NACT followed by radical surgery has been reported as a controversial alternative in locally advanced cervical cancer. The benefit of tumor downsizing regarding prognosis has not been proven (grade C) [10]. NCCN (National Comprehensive Cancer Network) guidelines have suggested that, although neoadjuvant chemotherapy followed by surgery has been used in areas where radiotherapy is not available, the data suggest no improvement in OS when compared with surgery alone for early-stage cervical cancer or locally advanced cervical cancer. A meta-analysis of data on patients with stage IB1 to IIA cervical cancer found that neoadjuvant chemotherapy may reduce the need for adjuvant radiotherapy by decreasing tumor size and metastases but indicated no OS benefit. However, data from a second meta-analysis suggested that the response to neoadjuvant chemotherapy was a strong prognostic factor for DFS and OS [6]. 

The response to neoadjuvant chemotherapy is the main factor predicting survival outcomes and the need for further treatments. Patients not responding to chemotherapy are characterized by the need for having adjunctive postoperative radiation therapy and poor prognosis. Hence, one of the main issues related to neoadjuvant chemotherapy is the inability to predict the response to chemotherapy. Having a practical tool able to predict the response to chemotherapy would be useful to select patients for receiving neoadjuvant chemotherapy or definitive chemoradiation. 

In patients with a diagnosis of cervical cancer, MRI is the gold standard to define initial staging, to monitor the response to treatment, and to evaluate recurrences.

Radiomics represents a translational field of research which consists of the extraction of data from standard radiological images from different imaging techniques (MRI, computed tomography—CT, ultrasound, positron emission tomography—PET, X-ray). The extracted data result in quantitative features describing the heterogeneity of the tumor and other intrinsic characteristics that may correlate with its biological behavior, response to treatment, and risk of recurrence.

The aim of our study was to develop a radiomics signature to predict the response to NACT in locally advanced cervical cancer using MRI images at the basal evaluation. 

## 2. Materials and Methods

This is a retrospective analysis conducted on patients enrolled in the study protocol 143-16 approved by the Institutional Review Board (IRB) of the Fondazione IRCCS Istituto Nazionale dei Tumori di Milano). Consecutive patients with squamous cell carcinoma, adenosquamous carcinoma, and adenocarcinoma of the cervix treated with neoadjuvant chemotherapy at the Gynaecologic Oncology Unit of the Fondazione IRCCS Istituto Nazionale dei Tumori di Milano between 2015 and 2020 have been considered. Patients signed informed consent forms and gave their permission for data and imaging collection for research purposes.

The inclusion criteria were as follows: (i) diagnosis of invasive cervical cancer; (ii) FIGO stage IB2-IIB cervical cancer at diagnosis; (iii) available basal magnetic resonance imaging. Exclusion criteria were as follows: (i) withdrawal of informed consent; (ii) lack of MRI images at baseline; (iii) neuroendocrine carcinoma. All the patients underwent MRI performed at the Fondazione IRCCS Istituto Nazionale dei Tumori at baseline as required by the study protocol and NACT was scheduled within 2 weeks.

All the patients, according to the study protocol, were treated with 3 cycles of neoadjuvant Carboplatin (AUC5) d1 and Paclitaxel (80 mg/m^2^) d1, 8, 15 q21 chemotherapy (dose-dense regimen). After 4 weeks from completion of NACT, patients were submitted to radical hysterectomy and pelvic lymphadenectomy. All the surgeries were performed at the Fondazione IRCCS Istituto Nazionale dei Tumori and all the histologic examinations were analyzed by a pathologist fully dedicated to gynecologic pathology.

For the study purpose, we compared two groups of patients based on their response to neoadjuvant chemotherapy: “Completely responding” and “Not completely responding”, according to the histopathological analysis of the uterine cervix at the time of planned radical surgery. In the “Completely responding” group we included patients with no residual tumor or microscopic (<3 mm) disease on the surgical specimen. Patients achieving partial response, stable disease, and progressive disease were grouped in the “Not completely responding” cohort. Of note, patients with pathological partial response (>3 mm) were grouped in the “Not completely responding” cohort since the goal of the study is to identify the cluster of patients who would benefit most from NACT. This image set was used for the training, cross-validation, and internal testing of 3 machine learning models. More specifically, in this work, (i) the segmentation of the VOI (volume of interest) was performed manually, slice by slice, by one expert examiner, using the Trace4Research segmentation tool (v1.0, DeepTrace Technologies, SRL, Milan, Italy) and MRI T2-weighted images. (ii) The pre-processing of image intensities within the segmented VOI included resampling to isotropic voxel spacing (1.5 mm). (iii) The radiomics features computed from the segmented VOI belonged to different families: morphology, intensity-based statistics, intensity histogram, gray level co-occurrence matrix (GLCM), gray level run length matrix (GLRLM), gray level size zone matrix (GLSZM), neighborhood gray tone difference matrix (NGTDM), neighboring gray level dependence matrix (NGLDM). Their definition, computation, and nomenclature are compliant with the IBSI (Image Biomarker Standardization Initiative) guidelines [13]. Steps from (ii) to (iii) were performed using the Trace4 Radiomics tool ((v1.0, DeepTrace Technologies, SRL, Milan, Italy). It must be noted that intensity histogram features were computed after an intensity discretization of the VOI, using a fixed bin width of 12. Texture features (GLCM, GLRLM, GLSZM, NGTDM, NGLDM) were computed after an intensity discretization of the VOI, using a fixed bin width of 12. Radiomic features were reported by Trace4Research according to IBSI standards. The convolutional layers of a pre-trained ResNet50, were used to extract a set of 2048 features from the images discretized using a fixed number of 256 bins and resampled to a dimension of 224 × 224 × 9 voxels. This family, called DeepFeatures, is not reported in the IBSI guidelines. Features with low variance (threshold = 0.1) were removed. Highly intercorrelated features were removed by a mutual information analysis (removing features with mutual information > 0.31). The selected radiomic features (informative and not redundant) were reported by Trace4Research according to IBSI standards. Steps from (ii) to (iv) were performed using the Trace4 Radiomics tool. Radiomic features were reported by Trace4Research. Three different models of machine learning classifiers were trained, validated, and tested for the binary classification task of interest (Not responding vs. Completely responding), based on supervised learning, using response to treatment as reference standard. For each model, a nested 6-fold cross validation method was used. The first model consisted of 3 ensembles of 36 random forest classifiers combined with Gini index with majority vote rule; the second model consisted of 3 ensembles of 36 support vector machines combined with principal components analysis and Fisher’s discriminant ratio with majority vote rule; the third ensemble consisted of 3 ensembles of 36 k-nearest neighbor classifiers combined with principal components analysis and Fisher’s discriminant ratio with majority vote rule. Oversampling technique for the minority class (Not responding) was applied by adaptive synthetic sampling method (ADASYN). The performances of the 3 models were measured across the 6 folds in terms of majority vote and mean area under the receiver operating characteristic curve (ROC-AUC), accuracy, sensitivity, specificity, positive predictive value (PPV), negative predictive value (NPV), and corresponding 95% confidence intervals (CI). The model with the best performance, according to ROC-AUC, was chosen as the best classification model for the binary task of interest (Not responding vs. Completely responding).

Statistical analysis was conducted with embedded tools of the Trace4Research platform. To describe the distribution of each of the most relevant features in the “Not responding” and “Completely responding” classes, we calculated their medians with 95% CI and presented graphically violin and box plots for intuitive visualization and interpretation. A non-parametric univariate Wilcoxon’s rank-sum test (Mann–Whitney U test) was performed for each of the relevant radiomic predictors to verify its significance in discriminating “Not completely responding” and “Completely responding” classes. To account for multiple comparisons, the *p*-values were adjusted using the Bonferroni–Holm method and the significance levels were set at 0.05 (*) and 0.005 (**).

## 3. Results

We collected MRI images from 72 patients meeting the inclusion criteria. Among those subjects, 28 patients (38.9%) belonged to the “Not completely responding” class and 44 patients (61.1%) belonged to the “Completely responding” class, according to response to treatment. The patients’ characteristics are presented in Table 1. Median age was 37 years old and the most represented histotype was squamous cell carcinoma (76% of patients). The Table 1 also shows the clinical and pathological differences between the two groups of patients: in the “Completely responding” group the median age of the patients was 43 years, and in the “Not completely responding” it was 46 years. In the “Completely responding” group, 35% of the patients had a FIGO stage IB disease and 28% had a stage IIB disease, while in the “Not completely responding group”, 18% of the patients had a stage IB disease and 34% a stage IIB disease.

### Radiomic-Based Machine Learning Modelling

From each segmented VOI of each image considered in this study, 3738 IBSI-compliant radiomic features were computed. Of these radiomics features, five resulted as being informative and not redundant (a variance above 0.1 and mutual information below 0.31). In particular, of the five selected radiomic predictors, two were intensity-based features (intensity histogram—logarithm filter—90th percentile, intensity histogram—Laplacian of Gaussian filter—robust mean absolute deviation), two were textural features (GLRLM—original image—Run entropy, GLCM—gradient filter—Correlation), and the last one was a deep learning-based feature (MR-T2W_DeepFeature317).

For the classification task of interest (28 images from the “Not completely responding” class vs. 44 images from the “Completely responding” class), these five predictors were used for the training, cross-validation, and internal testing (nested 6-fold cross validation) of three different models of machine learning classifiers considered in this work. Table 2A–C show the ROC-AUC, accuracy, sensitivity, specificity, PPV, and NPV as obtained from the training, cross-validation and internal testing of the three models consisting of three ensembles of machine learning classifiers. The ROC-AUC, accuracy, sensitivity, specificity, PPV, and NPV are reported with a 95% CI and *p*-value. Furthermore, for each model, the ROC curves for the three ensembles are plotted in Figure 1A–C. Based on the ROC-AUC, the model of random forest classifiers resulted as being the best model for the task of interest (28 images from the “Not completely responding” class vs. 44 images from the “Completely responding” class), with a ROC-AUC (mean) of 82% vs. the 67% of the support vector machine (SVM) model and 64% of the k-nearest neighbors (k-NN) model. The five selected radiomic predictors are shown in Table 3, together with their IBSI feature family and feature nomenclature. The predictors are ranked according to their statistical significance and to their frequencies among the selected predictors in the ensemble of the random forest classifiers. The median values of each feature, 95% CIs, and results from the univariate statistical rank-sum tests are also reported with the adjusted *p*-values. Furthermore, the two most relevant radiomic predictors (Intensity histogram—logarithm filter—90th percentile, Intensity histogram—Laplacian of Gaussian filter—Robust mean absolute deviation) also resulted as being statistically significant in discriminating the “Not completely responding” and “Completely responding” classes (the Bonferroni-corrected *p*-values < 0.05), according to the univariate statistical rank-sum tests. The violin plot and boxplot of the radiomic predictors are shown in Figure 2.

## 4. Discussion

In the present study, we developed and internally validated a diagnostic model based on machine learning and radiomics applied to baseline MRI images that can predict the response to NACT in patients with locally advanced cervical cancer. The baseline MRI images were processed for radiomic analysis and stable, non-redundant features were combined to create, train, and internally validate different models to classify the patients into the “Completely responding” and “Not completely responding” classes.

The best model developed (ensembles of random forest classifiers, Table 2A) showed an ROC-AUC (%) of 83 (majority vote), 82 (mean) [79.9–84.6], an accuracy (%) of 74, 74 [72.1–76.1], a sensitivity (%) of 71, 74 [68.7–78.9], and a specificity (%) of 75, 74 [71–77.5].

Although our results are preliminary, they represent an important step forward for the adoption of radiomics and machine learning models for cervical cancer patients.

As mentioned previously, NACT is not the mainstay of treatment for locally advanced cervical cancer patients, but it may be a treatment option in patients who cannot access radiotherapy in a reasonable time and in young patients who want to preserve sexual function or ovarian function [14]. In this pilot study, patients were divided into only two groups (“Completely responding and “Not completely responding”) by eliminating the gray area of partial responses and considering as responders to treatment only those patients who did not require further treatment after the scheduled radical surgery. A limitation of NACT is indeed the possibility for patients to undergo adjuvant treatments after radical surgery with a massive increase in morbidity, and the possibility of identifying chemo-responsive women as early as at diagnosis could allow for new scenarios and new possibilities for personalized care.

The advantages of our study were the consideration of the histological specimen after radical surgery as the gold standard for defining response and developing a model for the different tumor histotypes in accordance with the IBSI guidelines [13].

The limitations of the study were the small sample size and the consequent biases of the retrospective, single center study design. An important limitation is represented by the restricted replicability of the patient’s setting, since as previously discussed, NACT does not represent the standard of treatment in patients with locally advanced cervical cancer and therefore is not a therapeutic strategy adopted by the majority of centers.

There are now many papers in the literature that consider radiomics in the diagnostic–therapeutic process of the gynecological cancers. Our group has previously published several studies of radiomics and machine learning use in the triage of adnexal masses or in the differential diagnosis between uterine myomas and sarcomas [15,16,17].

Regarding the treatment of cervical cancer, there are many published studies, especially focusing on the Asian population. Tian et al. proposed a radiomic signature to predict the response to NACT using CT scans, which, however, is not the imaging method of choice for the patient’s workup and staging [18]. Aerts proposed that radiomics can quantify tumor heterogeneity by reflecting the different gene expression underlying the different responses to treatment [19]. Autorino et al. studied the development of a radiomics model based on MRI images to predict the long-term (2-year) survival in patients undergoing NACT therapy followed by radical surgery [20]. Other research has evaluated the use of radiomics applied to MRI and 18-FDG-PET in predicting the response to exclusive chemoradiation (standard treatment) [21,22]; in particular, some experiences propose the mean ADC value extracted from the baseline DWI MRI as an independent predictor of the disease-free survival in cervical cancer patients [23].

Although accumulating data are available in this setting, to the best of our knowledge, our study represents the first study of the use of baseline MRI images to extract the radiomic features related to the response to NACT in locally advanced cervical cancer. The results of our study are certainly encouraging, but we need external multicenter and prospective validation studies in a larger cohort of patients. Interestingly, radiomics represents a suitable tool for improving the characterization of different tumors and promotes a more personalized approach [24,25,26,27,28,29].

## 5. Conclusions

In conclusion, radiomics and machine learning can be applied to preoperative MRI images to select the most appropriate treatment in patients affected by locally advanced cervical cancer. Further studies are necessary to validate (externally) our results. More importantly, large prospective studies are needed to better understand the role of artificial intelligence and deep learning in this setting. At present, radiomic-based artificial intelligence seems a promising tool that might enhance the screening, early diagnosis, risk stratification, and prognostication in cancer patients. Further prospective evidence is warranted.

## Figures and Tables

**Figure 1 diagnostics-13-03139-f001:**
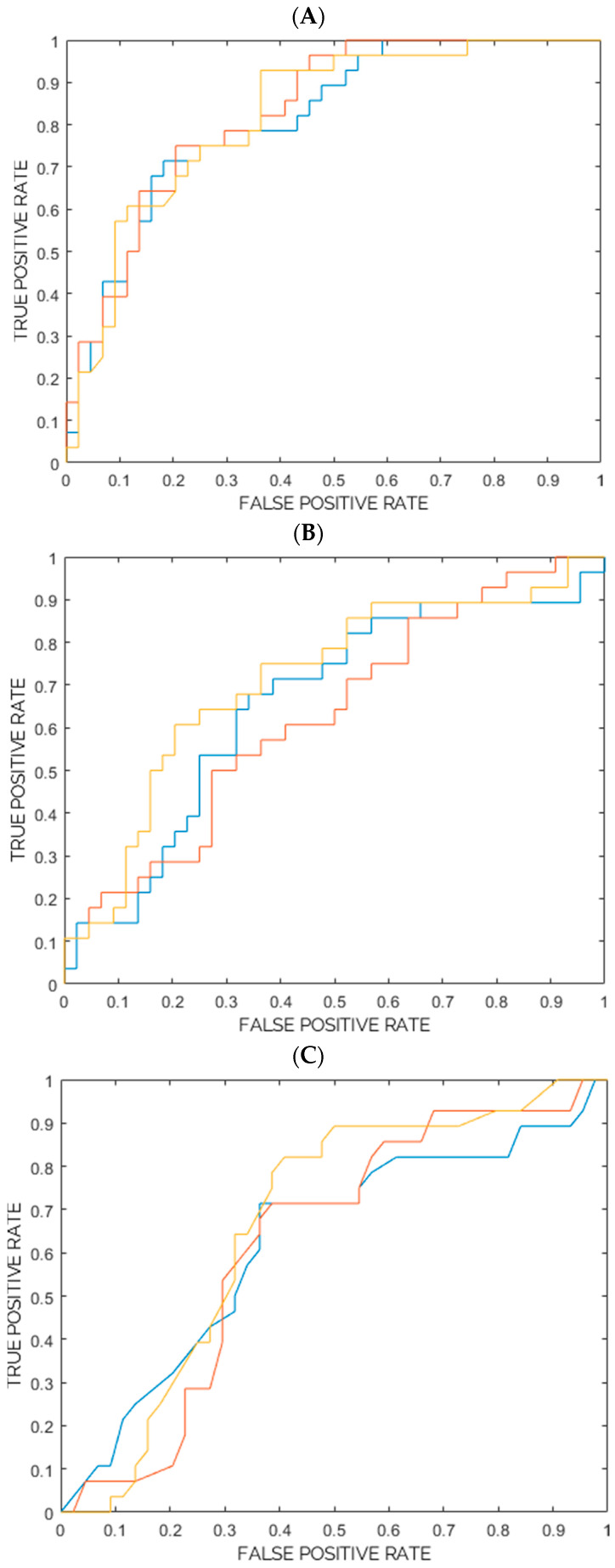
(**A**) ROC Curve for the model consisting of 3 ensembles of random forest classifiers (from internal testing). (**B**) ROC Curve for the model consisting of 3 ensembles of support vector machine classifiers (from internal testing). (**C**) ROC Curve for the model consisting of 3 ensembles of k-nearest neighbors classifiers (from internal testing).

**Figure 2 diagnostics-13-03139-f002:**
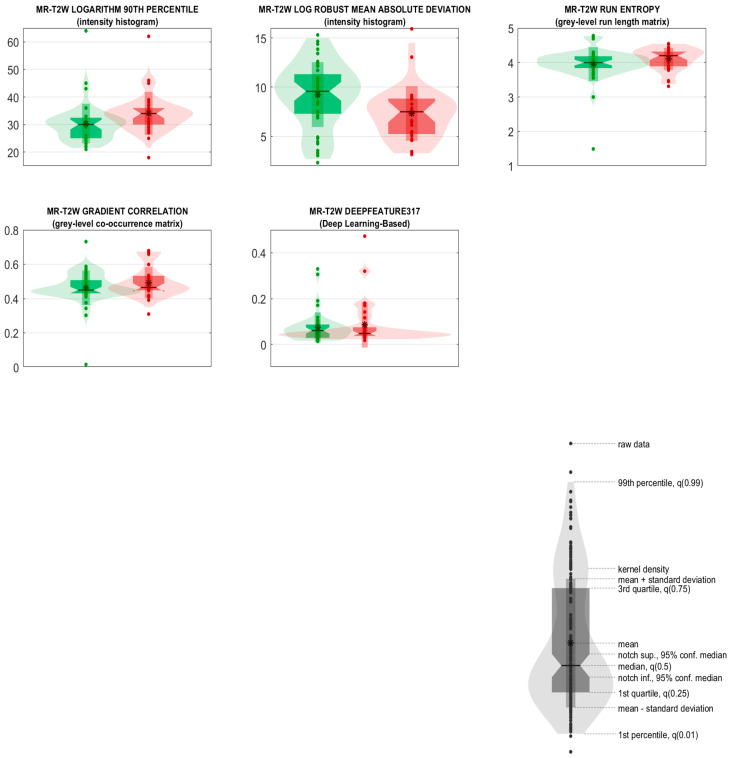
Ensemble of random forest. Violin and box plots of the radiomic predictors ranked from 1 to 5. Violin and box plots of Not responding and Completely responding classes are reported in red and green, respectively.

**Table 1 diagnostics-13-03139-t001:** Clinical and pathological characteristics of the patients included in the study. CR: “Completely responding” group, NCR: “Not completely responding” group.

Characteristics	72 pts	CR (28 pts)	NCR (44 pts)
Age (mean)	28–74 (37)	37–67 (43)	28–74 (46)
Histology			
Squamous cell carcinoma	55 (76%)	18 (64%)	32 (73%)
Adenocarcinoma	13 (18%)	8 (29%)	10 (23%)
Adenosquamous	4 (6%)	2 (7%)	2 (4%)
FIGO stage			
IB2	18 (25%)	10 (35%)	8 (18%)
IB3	15 (21%)	4 (14%)	11 (25%)
IIA1	8 (11%)	3 (11%)	5 (11.5%)
IIA2	8 (11%)	3 (11%)	5 (11.5%)
IIB	23 (32%)	8 (28%)	15 (34%)
Pathological response			
pR0	28 (39%)		
pR1–pR2	44 (61%)		

**Table 2 diagnostics-13-03139-t002:** (A) Model of 3 ensembles of random forest classifiers. (B) Model of 3 ensembles of support vector machine classifiers. (C) Model of 3 ensembles of k-nearest neighbors classifiers. Classification performance in terms of AUC, accuracy, sensitivity, specificity, PPV, NPV, corresponding 95% confidence interval, and statistical significance with respect to chance/random classification (*p*-value). Performance is reported for training, validation, and internal testing.

		Training	Validation	Internal Testing (Mean)	Internal Testing (Majority Vote—50% Threshold)
(A) Random forest classifiers					
	ROC-AUC (%) [95% CI]	100 * [99–100]	83 ** [79–87]	82 ** [80–85]	83
	Accuracy (%) [95% CI]	100 * [99–100]	75 ** [71–78]	74 ** [72–76]	74
	Sensitivity (%) [95% CI]	100 * [99–100]	75 ** [69–80]	74 ** [69–79]	71
	Specificity (%) [95% CI]	100 * [99–100]	75 ** [69–80]	74 ** [71–78]	75
	PPV (%) [95% CI]	100 * [99–100]	68 ** [64–71]	65 ** [62–67]	65
	NPV (%) [95% CI]	100 * [99–100]	84 ** [82–85]	82 ** [79–84]	80
(B) Support vector machines classifiers					
	ROC-AUC (%) [95% CI]	74 ** [73–75]	69 ** [65–72]	67 ** [56–77]	67
	Accuracy (%) [95% CI]	69 ** [66–71]	63 ** [63–64]	63 ** [53–73]	64
	Sensitivity (%) [95% CI]	68 ** [64–73]	64 ** [61–66]	63 ** [50–77]	64
	Specificity (%) [95% CI]	69 ** [68–70]	63 ** [61–65]	63 ** [54–72]	64
	PPV (%) [95% CI]	69 ** [66–71]	55 ** [54–56]	52 ** [41–63]	53
	NPV (%) [95% CI]	69 ** [66–71]	75 ** [73–77]	73 ** [63–82]	74
(C) K-nearest neighbors classifiers					
	ROC-AUC (%) [95% CI]	85 ** [83–86]	55 ** [48–61]	64 ** [57–70]	65
	Accuracy (%) [95% CI]	77 ** [75–80]	53 ** [46–61]	65 ** [54–76]	67
	Sensitivity (%) [95% CI]	84 ** [81–86]	56 ** [47–66]	75 * [60–90]	75
	Specificity (%) [95% CI]	71 ** [67–75]	51 ** [45–58]	58 ** [44–73]	61
	PPV (%) [95% CI]	75 ** [72–77]	44 ** [37–50]	53 ** [43–64]	55
	NPV (%) [95% CI]	82 ** [79–84]	65 ** [56–75]	79 ** [66–91]	79

* *p*-value < 0.05/** *p*-value < 0.005.

**Table 3 diagnostics-13-03139-t003:** Ensemble of random forest classifiers. The 5 predictors sorted in descending order according to their statistical significance and relevance.

#	Feature Family	Feature Nomenclature	Median in the Malignant Class (95% CI)	Median in the Benign Class (95% CI)	Uncorrected *p*-Value	Corrected *p*-Value
1	Intensity Histogram	MR-T2W_logarithm_90th Percentile	34 [32.22–35.78]	30 [28.22–31.78]	<0.005	<0.05
2	Intensity Histogram	MR-T2W_LoG_robust Mean Absolute Deviation	7.5 [6.43–8.57]	9.58 [8.62–10.53]	<0.005	<0.05
3	Gray Level Run Length Matrix	MR-T2W_Run Entropy	4.2 [4.07–4.33]	4 [3.92–4.08]	<0.05	0.19
4	Gray Level Co-Occurrence Matrix	MR-T2W_gradient_correlation	0.46 [0.44–0.49]	0.45 [0.43–0.47]	0.26	1
5	Deep Learning-Based	MR-T2W_DeepFeature317	4.75 × 10^−2^ [3.62 × 10^−2^–5.88 × 10^−2^]	6.03 × 10^−2^ [4.64 × 10^−2^–7.41 × 10^−2^]	0.66	1

## Data Availability

Raw data extraction will be provided by the first author (Chiappa V) upon reasonable request.

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
