# Peer review of "Using Radiomics and Machine Learning Applied to MRI to Predict Response to Neoadjuvant Chemotherapy in Locally Advanced Cervical Cancer"

_diagnostics, 2023, doi:10.3390/diagnostics13193139_

Round 1
Reviewer 1 Report
The manuscript is good, in general, it is a relevant, innovative work. The manuscript is understandable; It has few editing errors (which will be mentioned later). This work can be very important for the scientific community and useful for the clinical area; especially for those who work to combat the incidence of cervical cancer. The results presented are clear; and they contrast, for example in the images of the ROC curves, the evident differences between the machine learning models. This allows the authors to say that the random forest classifiers (AI tool) are the best tool to predict, from MRI, the most favorable response to Neoadjuvant chemotherapy NACT), in patients with locally advanced cervical cancer. Also, the tables presented here show the best values obtained from the tests, regarding training, validation and internal testing. which, together with the other results, allows them to indicate that the methodology proposed here can be very important to help predict the response to NACT in this type of patient.
On the other hand, the same authors pertinently mention the advantages and limitations of their study. Pointing out that the fact of using histological specimens after radical surgery is an advantage for this work, since they are the gold standard for defining response; and that an important limitation is the limited sample size. However, they suggest, this could be a work that allows other authors, with larger prospective studies, involving a better understanding of the role of artificial intelligence for screening, early diagnosis, risk stratification, and prognostication in cancer patients.
Finally, I would suggest minimal changes: for example, in the edition of the final paper, present the images larger (not as they are now on page 10). On page 3, there seems to be an error in the acronym DFS, because at the beginning it is said correctly, when talking about Disease Free Survival, but on that page it is as PFS. Then on page 13, there are some brackets, without the number that corresponds to the references that they want to highlight there. The initials VOI, I don't remember having seen them indicate what they refer to, and excuse my ignorance, that is the part that I couldn't understand in the manuscript. I would suggest the manuscript be accepted, with minimal changes, as long as the other referees and the editor also appreciate it.
Author Response
We thank the Reviewer for his comments.
We have made the minimal changes required to the manuscript in accordance with his suggestions.
- On page 3 we corrected the acronym PFS in DFS
- On page 13 we added the missing reference
- We explained the acronym VOI (Volume of Interest)
Reviewer 2 Report
This retrospective study aimed to use radiomics and machine learning at MRI to predict the response to neoadjuvant chemotherapy (NACT) in locally advanced cervical cancer. The study provides valuable method for predicting CR of patients with cervical cancer, but the following questions should be taken into consideration:
1.The details of the study should be given. For example, the interval between MRI imaging and NACT, and the interval between surgery and NACT? et al.
2.The discussion section should analyze the results and describe the clinical significance of the study systematically and comprehensively.
3.The section outlining the patient's basic information lacks the respective clinical and pathological features of the two groups.
4.The results shown in Table 2 and Table 3 should be described elaborately, specifically in the following aspects: the statistical significance of 5 radiomics predictors, the relationship between 5 radiomics predictors and 3 models.et al.
Moderate editing of English language required.
Author Response
Thank you for taking the time to review our manuscript.
Please see the attachment.
Best regards,
Valentina Chiappa

Reviewer 3 Report
''Using radiomics and machine learning at MRI to predict response to neoadjuvant chemotherapy in locally advanced cervical cancer''
The manuscript presented for review consists of 19 pages with 30 references. 5 tables and 4 figures are included. The paper is difficult to read and not formatted properly. The authors should follow the correct formatting for this journal. The manuscript is divided into 4 sections (Introduction, Material and Methods, Results, Discussion). Keywords are adequate. Moderate English changes are required (grammar, sentence structures). I recommend to check the article by a native speaker.
Abstract:
-The structure is clear (background, methods, results and conclusions). The aim is defined.
-FIGURE 2 | Ensemble of random forest. - Is not readable.
-Line 51: Could you explain abbreviations?
-Line 71: Could you explain abbreviations?
Line 217-222: I would recommend to separate these lines and create a new chapter - ''Conclusions''.
References:
-References are related to the issue and up-to-date.
-Considering the authors guidelines please check carefully and correct the punctuation of all references.
style, grammar structures, repetition of words...
-Line 44 : ''Countries'' - small letter.
Author Response

(The authors gave the same response as above.)

Round 2
Reviewer 3 Report
The Authors have included all suggestions.
I would accept the study in the present form.